# Esports for Seniors: Acute Effects of Esports Gaming in the Community on the Emotional State and Heart Rate among Japanese Older Adults

**DOI:** 10.3390/ijerph191811683

**Published:** 2022-09-16

**Authors:** Togo Onishi, Masayuki Yamasaki, Taketaka Hara, Tetsuya Hirotomi, Ryo Miyazaki

**Affiliations:** 1Graduate School of Human and Social Sciences, Shimane University, Matsue 690-8504, Japan; 2Faculty of Human Sciences, Shimane University, Matsue 690-8504, Japan; 3Faculty of Education, Shimane University, Matsue 690-8504, Japan; 4Interdisciplinary Faculty of Science and Engineering, Shimane University, Matsue 690-8504, Japan

**Keywords:** digital health, video games, heart rate, profile of mood states, esports

## Abstract

In the last few years, esports have become popular among older individuals. Although participation in esports can become a novel activity for older adults, evidence on their effects is limited to young individuals. This study investigated the effects of esports participation on the emotional and physiological states of older adults. Twenty-five older men and women participating in health promotion activities in community centers (75 ± 8 years old) played a two-player racing game (Gran Turismo Sport, Sony) for 8–10 min. Their heart rates (HRs) were measured while the subjects played the games. The blood pressure (BP) and Profile of Mood States (POMS) were measured before and after gaming. The average HR during games (98 ± 17 bpm) was significantly higher than at pre-gaming (76 ± 11 bpm, *p* < 0.001). The BP before and after the games did not significantly change. Interestingly, the vigor scores (positive mood affect) in POMS increased after the games (*p* < 0.05) in females, but not in males. In summary, among older individuals, playing esports games showed a moderate increase in HR, no spike/drop of BP, and positive mood elevation. Our preliminary data suggest the feasibility of participating in esports in a community for older adults and that it could affect mood positively, especially among older women.

## 1. Introduction

Esports, generally defined as “competitive video games” [1,2], have spread rapidly around the world in the last decade, mainly among young individuals. It is estimated that its market size in 2024 will exceed USD 1.6 billion, and the number of esports viewers will reach 577 million [3]. With this increased attention worldwide, significant research has recently been conducted to elucidate the scientific benefits of esports. Unfortunately, to date, the vast majority of previous studies have been cross-sectional epidemiological studies investigating associations between the playing habits and health parameters of esports. For instance, young esports athletes have been reported to be physically less active [4], more obese [5], and to have a higher prevalence of sleep disorders [6] in comparison with the general young population. Given these negative findings and potential health concerns, the recent massive boom of esports should be discussed carefully.

Studies on the physiological effects of esports are limited. In comparison with traditional sports, esports are considered to be a sedentary task, showing a similar energy expenditure to that of a resting state [7]. Despite its sedentary nature, previous studies have shown that a single esports session could have positive effects on physiological parameters. It has been well documented that the heart rate (HR) of esports participants is increased during games [8,9,10,11]. Previous experimental studies that analyzed young individuals reported that a single esports session was associated with faster response times [8], physiological arousal (moderate increase in stress hormones (e.g., cortisol)) [12], as well as the activation of sympathetic activity [9]. Moreover, Zimmer et al. (2022), as described above, reported that HR and blood cortisol levels reached their nadir 10 min after an esports session [7]. These physiological findings suggest that esports cannot provide the same metabolic reactions as traditional sports, but they could not trigger a negative stress response. Further widespread studies investigating both physical and mental parameters are needed in order to promote esports from a healthy viewpoint. However, evidence regarding the effects of esports remains scarce.

More crucially, evidence of the effects of esports has generally been limited to young populations. All of the above-mentioned studies, regardless of whether they employed an experimental [8,9,10,11,12,13] or epidemiological [4,5,14] approach, were conducted for young individuals, including young students [5,8,10,11] or young esports athletes in their 20 s to 30 s [4,9,12,13,14]. Very recently, esports have become popular among older populations [15]. In 2019, a world esports championship for seniors was held for the first time in history. Since then, several senior esports teams have been founded all around the world, for example in Sweden [16], as well as in Japan [17]. These developments have dramatically increased attention to “senior esports” and suggest the need to better understand the effects of esports gaming in older individuals. For instance, it is uncertain whether the increase in HR during esports is acceptable for older individuals’ health. Moreover, a previous study among young individuals [13] showed that acute esports activated sympathetic activity, but also increased stress and mental fatigue [13]. Given that older individuals are more frail than young individuals, these undesirable physical responses caused by esports need to be carefully considered.

Focusing on daily life use, esports could become a novel form of entertainment for older people. In recent years, older individuals have become accustomed to digital devices. Esports, which involve games that are designed to be fun to play, can therefore preferably affect the state of mind of healthy older individuals. In addition to physical factors, ageing is widely associated with psychological and behavioral changes. Older individuals are at high risk for emotional disorders, such as depression, which may result in functional disability [18]. Because a single video game session has been shown to enhance mood states among young individuals [19], it is expected that playing esports could elicit a positive mood among older people. However, no esports studies have investigated these effects in older populations. Obtaining an understanding of the potential physiological and mood responses of esports (as well as their interaction) among older individuals would help provide an understanding of appropriate “senior esports”.

Moreover, it is of note that older people generally play esports for recreation, not for competition. For instance, in Japan, an esports-specific facility for community-dwelling older people was founded in 2020 [20]. Therefore, in older populations, it is important to investigate the effects of esports under “real-life” environments, not under laboratory settings. Older people are known to enjoy communication with individuals in their neighborhood (such as close friends) and do not enjoy competition as much as young individuals [21]. As such, esports appear to be a suitable substitute for traditional forms of community-based activities. In this sense, previous findings from studies regarding esports, which were conducted for young individuals, do not seem to fit older populations.

Despite the increased social demand for “senior esports” worldwide, no scientific reports have been available, and the potential impact of esports on older adults has remained unknown. Investigations are warranted to understand how esports affect the mood and physiological states of older individuals. Therefore, the purpose of this preliminary study was to investigate the effects of participation in esports on mood and physiological responses among community-dwelling older men and women, under community-based settings.

## 2. Materials and Methods

### 2.1. Participants

To understand the acute effects of esports gaming among older adults in “real-life” environments, we conducted exploratory research as part of community-based health promotion activities. Community-dwelling older men and women (age ≥ 60 years old) were recruited from health promotion activities (light exercise, such as stretching and walking) in community centers (approximately 120 members in total) in two areas of Shimane Prefecture (Matsue City and Tsuwano Town), Japan. Although these health promotion activities were held by the community, the members and our university have previously collaborated for scientific purposes, such as through health lectures and medical measurements led by university staff. We recruited subjects using a flyer that was distributed to participants during their regular activities. All participants had been participating in the activities for at least a few months. Therefore, the participants in the present study usually met each other. An a priori power analysis was conducted with G*POWER 3.1 (Universitat Kiel, Kiel, Germany). Based on the means and standard deviations (SDs) reported in a previous study [8] using a two-tailed test with an effect size d of 0.64 and alpha error probability of 0.05, a total of 22 participants were needed for an achieved power (1-beta error probability) of 0.80 in order to detect differences in the HR while playing esports games. All of the subjects gave their informed consent for inclusion before they participated in the study. The study was conducted in accordance with the Declaration of Helsinki, and the protocol was approved by the ethical committee of the Faculty of Human Sciences of Shimane University approved the present study (2018-23). Moreover, prior to the gaming experiments, all of the participants underwent medical measurements (standard fitness measurement including anthropometrics, grip strength, walking speed, and resting blood pressure) and completed a questionnaire that included items related to their health history and exercise history. All participants were free from any cardiovascular disease according to their health histories and were living independently.

### 2.2. Study Protocol (Figure 1 and Figure 2)

This study investigated the acute effects of esports gaming on multiple outcomes. The outcome measures, including both mood and physiological measures, were examined before and after a session of esports gaming. All of the experiments were conducted in the community centers usually used by the participants. The study protocol is shown in Figure 1. All of the participants were fitted with a Polar OH1 (Polar Electro Inc.; Bethpage, NY, USA) heart rate monitor, which was attached to the participants’ arms throughout the study. For all participants, the heart rate (HR) was continuously sent to Polar Team (Polar Electro Inc.; Bethpage, NY, USA) using a tablet device. After at least 10 min in a seated position, each participant’s blood pressure was measured and their Profile of Mood States (POMS) was evaluated (pre-test). Thereafter, participants were allocated with age-matched pairs. The assigned pair sat in front of a screen (125 × 221 cm) and tested the game for 5–10 min for familiarization. After familiarization, participants played a game (Gran Turismo Sport, PlayStation 4, Sony, Tokyo, Japan), as described below. The screen was projected so that audience in the venue could see it. The rest of the participants could watch the pair’s game as an audience. The audience was allowed to give vocal support to the gaming pair. During the game, we measured the participants’ HR every 2 min, until the game ended. Within 5 min after the game ended, we collected the blood pressure and POMS (post-test).

Participants played the racing game Gran Turismo Sport. Each participant sat in a chair and played the game using a Logitech G29 (Logitech, Lausanne, Switzerland), which is a steering wheel-type controller. In the game, participants drove the N300 vehicle class on the “Northern Isle Speedway” track. In the race, the number of laps was set as 15. Because this game was played using steering wheel-type controllers, the participants were able to play without specific gaming skills. The play time depended on the content of the game, but generally the time per game was approximately 8–10 min. A representative photograph of a participant is shown in Figure 2.

**Figure 1 ijerph-19-11683-f001:**
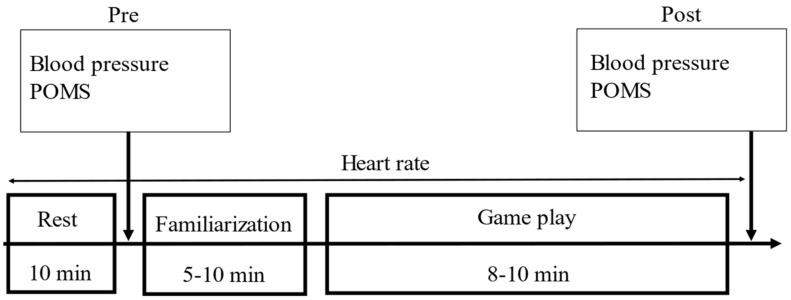
Study protocol.

**Figure 2 ijerph-19-11683-f002:**
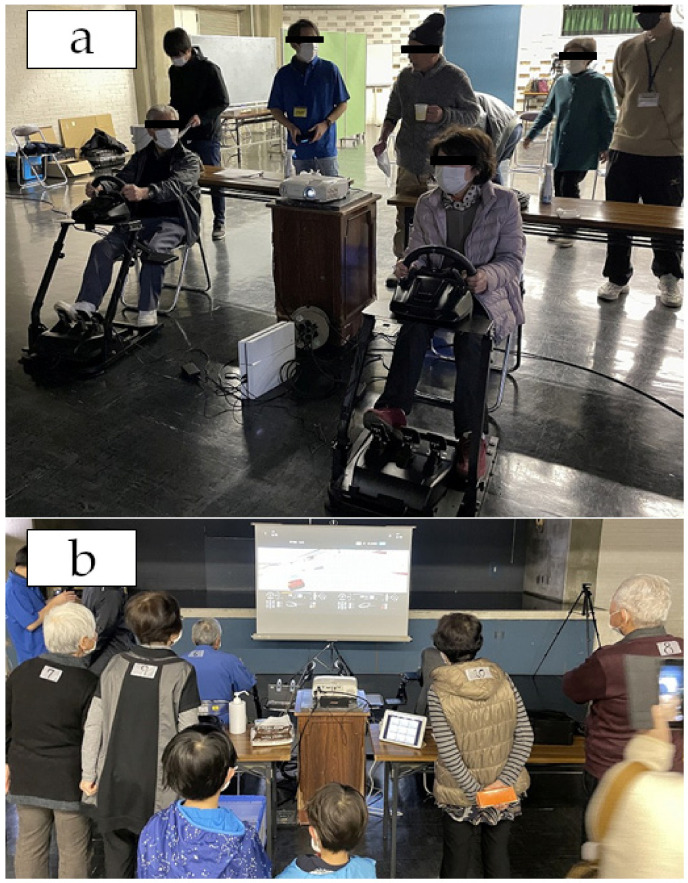
(**a**,**b**) Participants playing the game.

### 2.3. Profile of Mood States (POMS)

The profile of mood states (POMS) [22] assessment provides a rapid, efficient method for assessing transient, fluctuating active mood states, and is widely used for mood and mental health assessments. In this study, we used the Japanese version [23] of the short version of POMS2 [24]. POMS is composed of six scales: anger (A), confusion (C), depression (D), fatigue (F), tension (T), and vigor (V). The full version of POMS consists of a collection of 65 self-rated items that are used to assess the respondent’s feelings and emotions (e.g., “worn-out” or “active”), allowing for a quick assessment of the respondent’s mood state. Items are scored from 0 (“not at all”) to 4 (“extremely”). Experimenters need to clearly specify the time period in which the respondents should consider when evaluating the items (e.g., “past week, including today”, “right now” (employed in the current study)). Here, we used the short version of POMS, which consists of a select 30-item subset of the full version.

Higher POMS scores indicate greater levels of the corresponding construct. Note that only the V scale has a positive value, while all other scales have a negative connotation. We analyzed that the V subscale score as the positive mood score and the sum of the scores for the other five subscales as the negative mood score, according to the methods of previous studies [25,26]. The vigor score represents mental and physical energy levels [27]. For instance, vigor scores among athletes are generally higher that those among the controls [28]. The coefficients of reliability (Cronbach’s alpha) for the six scales were 0.57 to 0.88, which demonstrate their reliability [29].

### 2.4. Heart Rate and Blood Pressure

After at least 5 min of rest, the resting HR was measured using a heart rate monitor (Polar Electro Inc.; Bethpage, NY, USA). During the game, each participant’s HR was recorded every 2 min. HRmax was calculated using the age-predicted HRmax equation (220 − age) [30]. Before and after the games, blood pressure (systolic blood pressure and diastolic blood pressure) was measured (HCR-7501T, Omron, Kyoto, Japan).

### 2.5. Statistical Analysis

Given that most POMS measures showed a non-normal distribution (Shapiro–Wilk test: *p* < 0.05), nonparametric statistics were used, and intra-group comparisons were performed using Wilcoxon’s paired tests for paired samples. A paired *t*-test was used for each indicator to compare parameters before and after the games. The HR during the games was assessed using a repeated-measure analysis of variance with multiple comparison tests. Analyses were conducted using the SPSS 25.0 software program (IBM, New York, NY, USA). All data are shown as the mean ± standard deviation (SD) and *p* values of <0.05 were considered statistically significant.

## 3. Results

### 3.1. Participants

A total of 28 older men and women participated in the present investigation. No participants had esports gaming experience. After excluding participants whose gaming time was too short (<6 min) (*n* = 2) or participants with missing blood pressure data (*n* = 1), 25 participants (male, *n* = 9; female, *n* = 16; age, 75 ± 8 years [range, 62–94 years]; BMI, 23 ± 2 kg/m^2^) were eligible for the final analyses.

### 3.2. POMS (Table 1, Figure 3a,b)

Positive affect
Before and after the games, the vigor scores were significantly increased (*p* = 0.021). When stratified by sex, only female participants felt more vigorous (*p* = 0.022; Figure 3a), and this was not observed in males (*p* = 0.438, Figure 3b), demonstrating that only females experienced a more positive mood.

Negative affect
Esports gaming significantly increased the tension and fatigue scores (both *p* < 0.05; Table 1). When stratified by sex, although the fatigue scores were significantly increased in females (*p* = 0.013; Figure 3b), the summed negative mood scores did not change in males (*p* = 0.953) or females (*p* = 0.254).

**Table 1 ijerph-19-11683-t001:** Changes in POMS score (*n* = 25).

	Pre	Post	*p*-Value
Anger	2.2 ± 2.3	1.7 ± 2.6	0.180
Confusion	4.0 ± 2.7	4.3 ± 3.3	0.702
Depression	3.3 ± 3.0	3.0 ± 4.1	0.394
Fatigue	3.7 ± 3.6	5.8 ± 4.3	0.004
Tension	5.6 ± 3.6	7.2 ± 4.0	0.038
Vigor	11.1 ± 4.3	13.4 ± 3.6	0.021

Mean ± SD.

**Figure 3 ijerph-19-11683-f003:**
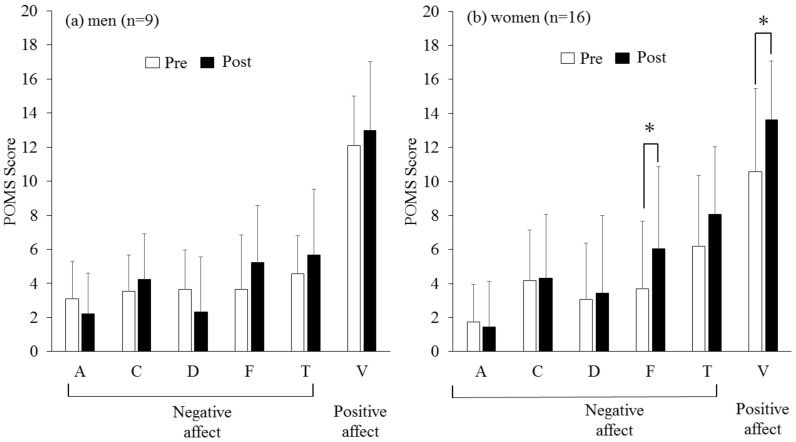
(**a,b**) Changes in POMS before and after esports games in men ((**a**), *n* = 9) and women ((**b**), *n* = 16). *: *p* < 0.05 vs. pre. Positive affect increased significantly in females (*p* = 0.022; Figure 3b), but not in males (*p* = 0.438, Figure 3a). Negative affect did not change in males (*p* = 0.953) or females (*p* = 0.254). A, anger; C, confusion; D, depression; F, fatigue; T, tension; V, vigor.

### 3.3. Heart Rate and Blood Pressure (Figure 4 and Figure 5, Table 2)

The mean HR (98 ± 17 bpm) and HR every 2 min during the games were significantly higher than the resting HR (76 ± 11 bpm, all *p* < 0.05, Figure 4 and Figure 5). The peak HR was 106 ± 19 bpm. The blood pressure before and after the games showed no significant change (systolic blood pressure 150 ± 24 mmHg vs. 153 ± 18 mmHg, and diastolic blood pressure 78 ± 13 mmHg vs. 80 ± 13 mmHg, Table 2).

**Figure 4 ijerph-19-11683-f004:**
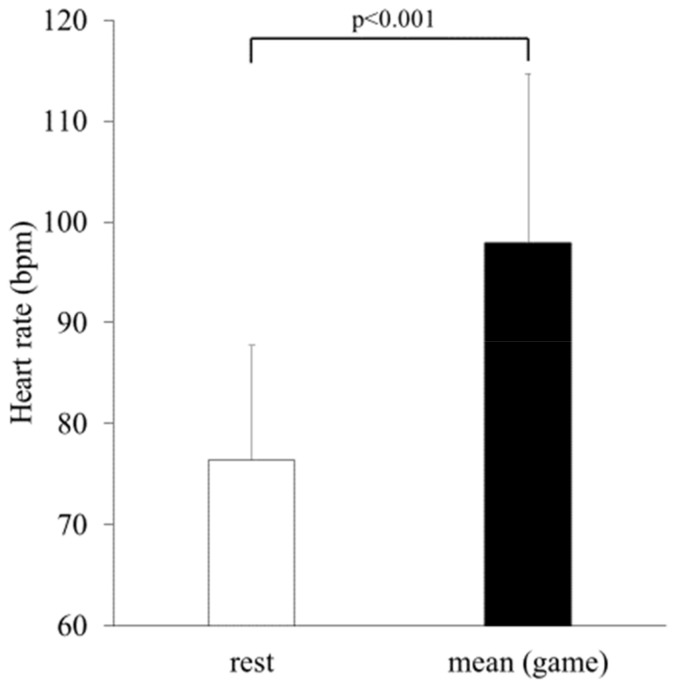
The rest and mean HR while playing esports (*n* = 25).

**Figure 5 ijerph-19-11683-f005:**
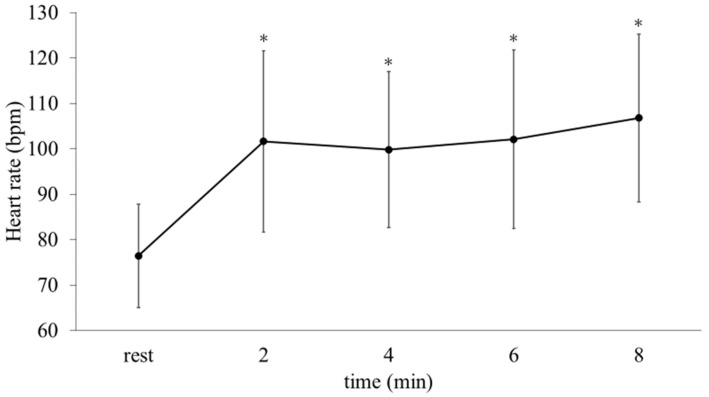
Changes in HR during the games (*n* = 25). *: *p* < 0.001 vs. rest. Note that the number of data points at 8 min was 16 because 9 participants had finished their games before 8 min.

**Table 2 ijerph-19-11683-t002:** Changes of blood pressure (*n* = 25).

	Pre	Post	*p*-Value
Systolic blood pressure (mmHg)	150.4 ± 23.7	153.4 ± 17.6	0.408
Diastolic blood pressure (mmHg)	78.3 ± 13.2	79.5 ± 13.3	0.475

Mean ± SD.

## 4. Discussion

The primary purpose of this preliminary study was to describe the response of mood and physiological indicators during esports gaming among community-dwelling older men and women in community-based situations. To the best of our knowledge, this was the first study to explore the effects of esports on older individuals. First, we observed that there were no adverse events (e.g., elevated blood pressure or undesirable increase of HR) among the older participants. Moreover, we found that playing esports was associated with a moderate increase in HR to 100–110 bpm. We also found sex differences in mood indicators. Only in older females, we found increased positive affect (vigor score) in POMS. Negative mood scores did not change in the males or females. In summary, these results suggest that participating in esports in the community could elicit positive affect and moderately increase HR; thus, esports might be suitable for improving mood, especially among older women.

The primary finding in the present study was the favorable changes in mood indicators. We found that only positive affect (vigor score) in POMS were significantly increased after games. This increased positive mood in our data are partly supported by previous studies. Kato et al. [19] reported using POMS scales, that playing several video games (three of five titles) increased vigor in POMS among young individuals [19]. Therefore, regardless of the game title, esports may increase vigor. However, unlike the above-mentioned study [19], we observed significant increases in parts of negative affect (tension and fatigue). This discrepancy could be related to the setting. While the participants in the study of Kato et al. [19] played the games alone, our participants played a one-on-one racing game. Supporting this, previous studies [31,32] have shown that playing video games against a friend is more exciting (e.g., induces a higher galvanic skin response) [31] in comparison with playing against a computer. 

Interestingly, only in females, we found increased positive affect. These results suggest that participation in esports may affect older people’s mood differently with sex. Our findings are partly supported by previous studies. Mackintosh et al. [33] reported that single Wii Boxing play affected mood in POMS among young men and women differently [33]. In their study, men experienced increased tension, while women experienced increased vigor [33]. The mechanisms of these sex-specific changes are unknown; however, this may be explained by both biological and social (environmental) factors. First, it is well known that biologically, men are more competition-oriented than women [34]. Supporting this, a previous video game study found that boys preferred competitive video games (two-player games), while girls preferred cooperative games [35]. Based on this biological evidence, our male participants would have played for victory rather than fun, thus their vigor did not change. In contrast, our female participants may have become more energetic because they played for fun rather than to win. Second, social (environmental) factors would have affected the results. Our participants were all very close friends. In our experimental sites, the players were surrounded by an audience (including other study participants) (Figure 2). In this situation, female participants tended to cheer for the players. It is well known that women prefer closer relationships than men. Taken together, unlike esports among young individuals (mainly played by men), when played with close friends, such as our participants, esports might be more suitable for improving mood among women than among men. However, because our findings were only based on a single playing session, the longitudinal changes remain unknown. This should be investigated in future studies.

In addition to positive affect, HR was significantly increased during the games. The magnitude of increase in HR levels in the present study is comparable to that observed in previous studies. In the present study, the peak HR was 106 ± 19 bpm (73% of their age-matched HR max) and the increase was approximately 30 bpm in comparison with the resting state. In previous studies, in which young esports players played a game (Fortnite) under daily life conditions, the peak HR during the game was 120 ± 16 bpm (60%HRmax) [11]. In contrast, the %HRmax in other studies among young amateur esports players under competitive conditions (such as esports competitions) were 81% [8] and 94% [10]. Considering this difference in HR response according to the gaming environments, it is plausible that not age, but gaming conditions, might be important for controlling HR response while playing esports.

Beyond the desirable mood and physiological changes, our data may also suggest that for older individuals, participation in esports with close friends was not associated with specific adverse events; thus, it could become a meaningful community activity. First, in this study, there was no significant change in blood pressure from before to after the game. Because a previous study among young individuals reported that blood pressure increased after tournaments [8], it seems that our community-based “real-life” experimental conditions may be acceptable for older people. Furthermore, in the present study, although the HR of older individuals showed a significant increase during games, the magnitude of increase in the mean HR (increase of 22 ± 12 bpm) from the resting state was equivalent to that during walking (increase of approximately 25–26 bpm) [36]. Taken together with the unchanged blood pressure levels and moderate HR increase, it is plausible that playing esports can become a community-based activity for older individuals. 

The present study was associated with some limitations. First, the study population was relatively small. Therefore, our findings regarding sex differences should be confirmed in larger study populations. For instance, the esports gaming experience would have influenced our results. In the present study, none of our participants had esports gaming experience. Therefore, our findings may have been influenced by their lack of experience. However, because the number of older individuals with esports gaming experience is thought to be limited, our participants were considered to be representative of general older individuals. Therefore, we think that our findings are clinically important. Second, our sampling period (<10 min) may have been too short to adequately observe the physiological responses. We decided to use this short period out of concern for the safety of our older and unexperienced participants. In fact, the sampling period in previous video game studies (active video games) among children was 10 min [37]. Furthermore, while we did not observe elevated blood pressure, a study has reported that resting blood pressure increased after a single esports session [8]. Therefore, our experimental design could be justified for older individuals. Third, we had no control group. Therefore, it is possible that if our participants had played other game titles, we might have observed different mood responses. Further studies should investigate the mood and physiological responses among older individuals using different game titles. Finally, our experiment was conducted under community-based real-life conditions. This was because older people would generally play esports as part of their recreational activities. However, to validate our findings, future studies should investigate the effects of esports in older individuals under more controlled laboratory settings.

## 5. Conclusions

We found that participation in esports caused a mild increase in HR with unchanged blood pressure, along with favorable changes in mood parameters (increased positive mood and unchanged negative affect). In addition, we also found an increased positive affect, mainly in females. These results suggest that community-based participation in esports seems feasible and may be suitable for improving mood in older adults, especially among older women. However, our study was a clinical study conducted under community-based real-life situations. Clearly, further studies are needed to generalize our findings, and to investigate physiological mechanisms that underlie our findings and to investigate the long-term effects.

## Data Availability

The dataset supporting the conclusions of this article is available upon request to the corresponding author.

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
