# Peer review of "Esports for Seniors: Acute Effects of Esports Gaming in the Community on the Emotional State and Heart Rate among Japanese Older Adults"

_ijerph, 2022, doi:10.3390/ijerph191811683_

Round 1
Reviewer 1 Report
Firstly, I would like to thank you for the opportunity to review the following manuscript. Esports really have spread rapidly in the last decade and new research is needed to see all it’s negative and also positive effects. Also, it’s benefit in older population should be investigated. This study is a good start. However, although the text is well written and it is easy to understand and this may become an interesting report from your research, in my opinion it still requires work on the academic soundness and there are some aspects that compromise the quality of this research and need to be improved. I hope with my comments you will be able to enhance the quality of your work.
Title: The title is too vogue. I feel this is not an adequate title, as the authors actually found also negative mood scores to increase as well. Please consider reformulating the title and keep it realistic and simple.
Abstract: line 20, here authors say that age is 76±8 and in the text 75±8.
In the next line authors say that Heart rate, resting blood pressure and the Profile of Mood States (POMS) were measured pre-, during, and post-gaming. That is not correct according to the protocol where only HR was measured during gaming. Please decide what information is correct.
Introduction: Authors use term preliminary study. What do you mean by this?
Material and methods:
Participants.
Authors should indicate the type of study design.
Has any predetermination of the sample size been made?
The sample size is small. If you sent flayers in two centers, you should have had better response. Did you try to go there by yourself and speak to potential participants? Response should be much higher.
What are serious diseases that you refer to? Can you please say what conditions participants had? And are there any inclusion/exclusion criteria? They should be indicated.
Study protocol
Authors say that the experiments were conducted as a part of health promotion activities in which participants had been participating for at least a few months. Can you tell me which were activities you are referring to and why is that important for your study? Is there any particular reason why they had to be pre-involved in those activities?
Pre test and gaming are clearly described. For pre test you had 10 min in seated position before you measured BP and that is minimum for resting values, but why did you choose 5 min after the game? You did not measure it at the end of the game according to this protocol. Can you please tell me why did you choose to measure BP after 5 minutes after the game and not at the end of the game or 10 min after?
Statistical analyses
Have normality tests been performed? Are there outliers?
Which test did you use when you divided the participant group into two according to the sex?
Results:
It is not clear what was the mean age of the participants, as mentioned before. Also if the oldest participant had 94 years can you check SD?
Did that person 94 years old had age matched opponent?
Authors mentioned only HR at rest and after game. But they also mentioned that HR was continuously measured (and in some section of the paper they mentioned it was measured every 2 minutes). Where are those results? Also, results of the heart rate variability would be useful.
Reviewer 2 Report
The authors investigated the effects of esports participation on the emotional and physiological states of older adults by measuring Heart rate (HR), resting blood pressure (BP), and Mood Profile (POMS) before, during, and after play.
The research topic and methods of the study are available in many types of esports for young individuals. Although it does not contain much innovation in terms of subject and methods; In particular, I think the study of mood and physiological responses of e-sports participation on older populations is invaluable.
Overall thorough study with a good rationale and methodological approach is presented, and the data are convincingly analyzed and shown. However, I have some suggestions. Authors are encouraged to make suggested changes and improvements before resubmission. Please find the below comments that should be considered while improving the paper:
1. Please remove the "Input:, Method:, Results: Conclusions:" in the Abstract section.
2. Please provide information about the Vigor scores (and ev. reference).
3. Please polish the manuscript, and eliminate the many mistakes in the original text.
For example,
While the p-value of the Vigor score in Table 1 is 0.021, the p-value in the 194th row is 0.029. Please fix them.
4. Please check the format of the references.
With best wishes
Reviewer
Round 2
Reviewer 1 Report
Authors made changes that were needed.